# Economics of a Feeding Budget: A Case of Diversity of Host Plants for *Lymantria dispar* L. (Lepidoptera) Feeding on Leaves and Needles

Vladislav Soukhovolsky [1,2,3], Olga Tarasova [2], Sergey Pavlushin [4], Ekaterina Osokina [5], Yuriy Akhanaev [4], Anton Kovalev [3,*] and Vyacheslav Martemyanov [4,*]

[1] V. N. Sukachev Institute of Forest SB RAS, Akademgorodok 50/28, 660036 Krasnoyarsk, Russia
[2] Department of Ecology and Nature Management, Siberian Federal University, Av. Svobodny 79, 660041 Krasnoyarsk, Russia
[3] Krasnoyarsk Scientific Center SB RAS, Akademgorodok 50, 660036 Krasnoyarsk, Russia
[4] Institute of Systematics and Ecology of Animals SB RAS, Frunze Str. 11, 630091 Novosibirsk, Russia
[5] Department of Natural Science, Novosibirsk State University, Pirogova Str. 1, 630090 Novosibirsk, Russia
* Correspondence: sunhi.prime@gmail.com (A.K.); martemyanov79@yahoo.com (V.M.); Tel.: +7-9039236335 (A.K.)

**Abstract:** Relationships were analyzed among the energy-related characteristics of feed consumption by caterpillars of the spongy moth (also known as gypsy moth) *Lymantria dispar* L., survival of individuals, and fecundity of females depending on the species of a host plant. An optimization model of feed consumption was used for the calculations. In this model, efficiency of consumption depends on two parameters: efficiency of metabolic degradation of feed and efficiency of caterpillar biomass synthesis. Experiments were conducted regarding the feeding of caterpillars on the leaves of silver birch *Betula pendula* Roth. and needles of Siberian larch *Larix sibirica* Ldb. and Scotch pine *Pinus sylvestris* L. On the basis of the results of experiments, "costs" of the feed for females and males were calculated, the consumption efficiency of different types of feed was found, and the relationship between efficiency of feed consumption and female fecundity was determined. The proposed approach can be employed to assess feeding efficiency of insects in various habitats.

**Keywords:** insect; nutrition; pine; larch; birch; energy balance; survival; fecundity; modeling

## 1. Introduction

Among the factors of natural regulation of population dynamics of insects (some of the main first-order consumers on the planet), food resources are especially important [1]. Processes of natural regulation of life curves are expected to exist mainly in natural terrestrial ecosystems, such as forests or meadows. Therefore, in the text below, we focus on the analysis of forest ecosystems as systems capable of self-regulation and occupying a significant proportion of the area on our planet. One of the most crucial factors influencing forest insect population dynamics is the fecundity of individuals [1]. Fecundity, in turn, is determined by the weight of adults (imago) and its strong correlates: weight of pupae or caterpillars of older ages before pupation [2]. This relation is especially well pronounced in species having nonfeeding imaginal stages (capital breeding animals), when the whole reserves of matter and energy are determined by accumulated reserves in juvenile stages. For instance, in females of spongy moth *Lymantria dispar* L. and Siberian moth *Dendrolimus sibiricus* Tschetv., the number of eggs depends linearly on the weight of older caterpillars [2]. As a consequence, there is a relationship between the amount of available food and weight of individuals. Nonetheless, the assessment of the amount of food resources amount—leaves or needles (for phyllophages) or cambium weight (for xylophages)—is not entirely correct because it is necessary to take into account not only quantity but also quality of food. For humans, the simplest indicator of food quality is the calorie content of foods, but no such

indicators for insects have been proposed. Nevertheless, the assessment of feed quality according to the energy balance of its consumption has been used by various researchers for a long time. The existence of relationships between the survival and fecundity of insects and the energy balance of nutrition has been confirmed by numerous experiments on the feeding (and weight gain) of caterpillars under laboratory conditions [3–7]. These dependences are critically important for polyphagous consumers that can feed on different species of plants. Due to this energy balance of nutrition, investigators can distinguish preferred, less preferred, and nonpreferred plants. Moreover, on the basis of the calculation of the energy balance for specific species of host plants, it will be possible to predict the prospects for adaptation to native plant species in the event of invasion/expansion of a geographic range of polyphagous phytophages.

In experiments on the feeding (and weight gain) of forest insect caterpillars, it is possible to estimate the balance of energy received by an individual with food. The balance equation takes into consideration expenditure of energy on the increase in the individual biomass, metabolic processes, and excretion with excrement [8–10].

$$E = M + R + H \tag{1}$$

where $E$ is energy received with consumed food, $M$ is increase in the individual's biomass, $R$ is metabolic expenditures, and $H$ is the energy excreted in excrement.

To describe the energy balance, various parameters are used, such as the utilization coefficient $UC = (E - H)/E$, efficiency of food assimilation $EFA = M/(E - H)$, and efficiency of the use of consumed food $EUC = M/E$, characterizing the ecological efficiency of nutrition of a phytophage [11]. Note that these three parameters of food consumption by insects are not independent, e.g., $EUC = UC \times EFA$. Having determined the values of $E$, $M$, and $H$ from experiments, it is possible to unambiguously calculate the metabolic costs (in mass units).

Mass balance calculations during feed consumption are widely used in the evaluation of insect growth. For example, such calculations are used when choosing optimal diet for the caterpillars of *Pieris brassicae* L. in the laboratory [12] or choosing optimal insect species for use as a transformed animal feed [13].

Nonetheless, these relationships alone do not completely determine patterns of food consumption by insect caterpillars. For a fixed value of $E$, the first balance equation contains two independent variables (any pair of three variables: $M$, $R$, and $H$). Therefore, there is no unambiguous solution, and food consumption parameters may vary.

In this work, a modified approach was used [14,15] to describe the energy balance during feed consumption by phyllophagous insects. This approach allows obtaining exact solutions of the balance equation for energy-related characteristics of feed consumption. In particular, "economic prospects" for caterpillars of the spongy moth were assessed if it massively moved to the north and started to consume coniferous species in boreal forests. This event was predicted by both modeling studies [16] and empirical data [17]. By means of a modified balance equation, for spongy moth caterpillars, characteristics of feed quality were determined when the caterpillars were fed with needles of Siberian larch *Larix sibirica* Ldb. or of Scotch pine *Pinus sylvestris* L. These plant species from the southern taiga can be consumed by the spongy moth according to previously published data [18]. In addition, we estimated the energy balance during feeding on the leaves of silver birch *Betula pendula* Roth. as the preferred species of a host plant for the studied spongy moth population. Because it is known that the feeding experience of the parental generation affects the feeding of subsequent generations in this species [19], we implemented intermediate adaptation to the plant species in question. This situation simulates an increase in the number of insects in biotopes where the preferred host plant is not dominant, e.g., in invasion areas. In the case of Siberian populations of spongy moth, the zone of broad-leaved forests (endemic zone) transitions into a zone of mixed forests and southern taiga (zone of recent invasion), where conifer trees begin to dominate [17]. Thus, with an increase in the number of consumers in the invasion zone, there is a high risk of switching to an alternative host plant because the preferred resource is in short supply. Moreover, this transition can be preserved in the

feeding pattern of daughter generations, owing to the induction of chemical defense and increasing antibiosis in a limited number of preferred host plant species [20,21]. In other words, we are still evaluating initial steps of consumer adaptation to alternative nutrition but not the first step.

## 2. Materials and Methods

### 2.1. Studied Species

For this study, the spongy moth *Lymantria dispar* L. (Lepidoptera: Erebidae) was used, which is one of the main defoliator species in temperate forests of the Holarctic [22–25]. This species is polyphagous and consumes more than 300 species of deciduous plants [26,27]. A number of studies involving pheromone monitoring have revealed a significant trend of a northward shift of geographic-range boundaries of this species in the central part of Eurasia [17] (Ponomarev et al., unpublished data). In the West Siberian part of its range, *L. dispar* mainly consumes *B. pendula* leaves. Judging by the structure of forest communities in newly invaded areas, we chose two species of conifers that can potentially be consumed by *L. dispar* caterpillars [18]: Siberian larch *Larix sibirica* Ledeb. and Scotch pine *Pinus sylvestris* L.

The insects were collected under natural conditions (a birch forest) in autumn 2020 as diapausing embryos in eggs. The northern boundary of sites of mass reproduction of the spongy moth in the central part of Eurasia was chosen—Vengerovo (N 55° 40′ 52.6584″ E 76° 45′ 6.0444″)—as the sampling site. In other words, this population has the highest chance of settling in the southern sub-taiga, especially if we consider the dominant winds and the ability of these caterpillars to parachute [28]. In the summer of 2021, hatching caterpillars (mixed offspring of 50 females) were divided into three groups of 100 insects and reared to the imago stage in hatcheries under laboratory conditions (24 °C, natural lighting, 40% humidity) on silver birch (*Betula pendula* Roth.), Siberian larch (*L. sibirica*), and/or Scotch pine (*P. sylvestris*). An evaluation of the viability and physiological state of the parental generation was presented in a study by Pavlushin et al. (manuscript submitted). From the parental generation, offspring were obtained in each group; these offspring after wintering were used for the present experiment in 2022.

In the spring of 2022, all laid eggs (stored in the winter at 2 °C) were taken out of diapause, and larval hatching was quantified. Prior to the experiment, all three groups of first-age insects were reared only on birch to prevent differences in the rate of larval development. All measurements were started after the larvae reached the second age. This approach can simulate the situation in nature when younger insects feed on a limited amount of preferred species (limited resource), and, after reaching larger sizes, they spread to other plants and feed on what they have to; under conditions of taiga and a zone of mixed forests, these are coniferous plants.

Each group of second-age insects was placed on its host plant, in accordance with the "parental diet": birch, larch, or pine. The insects were kept individually in 125 mL feed containers. Each feed group consisted of 30 insects. Small fresh twigs with birch leaves, larch needles, or young pine needles served as the feed. After calculations, the feed was provided with a small excess to completely prevent starvation. Feed was refreshed, and all parameters were recorded every 2 days. The following parameters were determined: weight of caterpillar, weight of fresh feed, weight of excrement, and weight of feed residues (leftovers of leaves and needles). The weighing was performed using an electronic scale with 1 mg accuracy. To measure natural evaporation of moisture from the feed (transpiration), a control subgroup was set up in each feed group. This subgroup was composed of identical containers with twigs of birch leaves (or needles) but without insects, and the measurements of feed weight were carried out at the same intervals, i.e., every 2 days. Then, we calculated the water loss in control leaves (i.e., over 2 days) and corrected (added) this related weight to bits of leaves/needles remained after larvae feeding. Constant humidity was maintained in the laboratory room. At the end of experiment, the weight of surviving

pupae was recorded, and the sex of each insect was determined. The weight of consumed feed was adjusted for water transpiration.

### 2.2. The Model of Energy Balance during Feed Consumption by the Insect Caterpillars

To describe patterns of feed consumption by phyllophagous insects, the authors of the article (V.S. and O.T.) proposed a model of optimal feed consumption [14,15]. In the model, the initial variables associated with energy in the process of analytical transformations are replaced by mass indicators. Further, the mass of components is used in formulas and calculations. As proposed in this model, Figure 1 shows the scheme of distribution of energy $E$ received by a caterpillar when consuming feed.

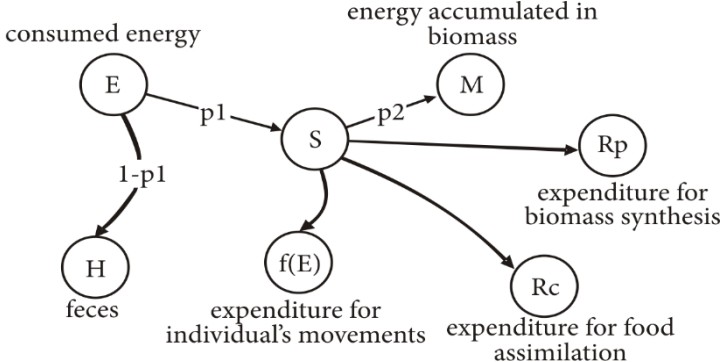

**Figure 1.** The scheme of distribution of energy $E$ received by a caterpillar when consuming feed.

The distribution of energy received with feed during growth of caterpillar is as follows: $H = (1 - p_1)E$ is the energy excreted with excrement, $S = p_1 E$ is the energy consumed by larva, $M = p_2 S = p_2 p_1 E$ is the energy transformed into caterpillar biomass (including exuvium), $R_p$ is the metabolic expenditure, $R_c$ is the expenditure on feed conversion, and $f(E)$ is the energy spent on movements of an individual in search of food.

Quantity $\frac{p_1 E}{E} = p_1$ is the UC (utilization coefficient), quantity $\frac{E_2}{E} = \frac{p_1 p_2 E}{E} = p_1 p_2$ is the EUC (efficiency of consumed feed use), and quantity $\frac{p_1 p_2 E}{p_1 E} = p_2$ is the EFA (efficiency of assimilated feed use).

In the model of optimal feed consumption, the balance equation for distribution of energy $E$ received by an individual with feed can be written as

$$E = (1 - p_1)E + a p_1^2 E + p_1 p_2 E + b p_1 p_2 + f(m, E) \tag{2}$$

where $(1 - p_1)E$ is the energy excreted in excrement, $a p_1^2 E$ is the expenditure on preparing feed and transforming it into a form suitable for consumption, $p_1 p_2 E$ is the energy transformed into biomass of an individual, $b p_1 p_2 E$ is the metabolic expenditure on the growth of an individual, and $f(m, E)$ is the expenditure on movements of the caterpillar in search of feed.

The difference between balance Equation (2) and traditional Equation (1) lies in the presentation of metabolic expenditures as two independent components: expenditure on feed preparation $a p_1^2 E$ and expenditure on actual metabolism $b p_1 p_2 E$.

In feeding experiments, caterpillars are kept in small cages and are supplied with feed regularly. Under these conditions, movements of caterpillar in search of food are minimized, and it can be assumed that $f(m, E) \to 0$. In this case, after simple algebraic transformations of Equation (2), we obtain

$$1 = a p_1 + (b + 1)p_2 \tag{3}$$

Equation (3) characterizes the energy balance of a larva in relative units. In the plane $\{p_1, p_2\}$, Equation (3) describes a straight line intersecting the abscissa axis at point $1/a$ and the ordinate axis at point $1/(b + 1)$. It can be said that the straight line described by Equation

(3) is the geometric locus of all values of $p_1$ and $p_2$ allowed by energy considerations. The straight line described by Equation (3) imposes restrictions on $p_1$ and $p_2$ but does not allow calculating their values because it is impossible to determine two unknowns in one equation. To determine these parameters, an additional equation is needed.

Suppose that feed consumption by insect caterpillars is optimal: during this process (all else being equal), maximum efficiency $q$ of feed consumption is ensured:

$$q = \frac{p_1 p_2 E}{E} = p_1 p_2 \rightarrow \max \tag{4}$$

The product $p_2 p_1$ is proportion of total energy $E$ spent on the synthesis of larva biomass. In the plane $\{p_1, p_2\}$, the geometric locus of all possible values of $p_1$ and $p_2$ is hyperbola $p_2 = \frac{q}{p_1}$. Combining energy and populational considerations, we can conclude that $p_1$ and $p_2$ values should ensure that product $p_1 p_2$ is maximal when the budgetary constraint (Equation (3)) on $p_1$ and $p_2$ is taken into account.

Quantity $q$ makes sense if the following conditions are met:

$$p_1 > 0; \; p_2 > 0 \tag{5}$$

From Equation (3), $p_2$ can be derived as follows:

$$p_2 = \frac{1}{b+1} \cdot (1 - a p_1) \tag{6}$$

Because $p_2 > 0$, the following condition should be true:

$$(1 - a p_1) > 0 \tag{7}$$

We can transform Equation (4) by means of Equation (6), assuming that Condition (7) is met:

$$q = p_1 p_2 = \frac{1}{b+1} \cdot p_1 (1 - a p_1) \rightarrow \max \tag{8}$$

Optimal values of $\hat{p}_1$ and $\hat{p}_2$ at which $q$ is maximal are easily calculated from conditions $\frac{dq}{dp_1} = 0$ and $\frac{d^2 q}{dp_1^2} < 0$:

$$\hat{p}_1 = \frac{1}{2a}; \hat{p}_2 = \frac{1}{2(b+1)} \tag{9}$$

If $p_1$ and $p_2$ values can be computed from the results of caterpillar feeding experiments, then it is possible to determine ecological cost $a = \frac{1}{2\hat{p}_1}$ of feed preparation and ecological cost $(b+1)$ of the synthesis of biomass of an individual: $b + 1 = \frac{1}{2\hat{p}_2}$ (Figure 2).

If the ecological cost of feed preparation is high, then a situation is possible where Condition (7) stops being true. Practically, this means that the cost of preparing feed exceeds the energy coming with feed. If Condition (7) is not fulfilled, it leads (as shown below) to the death of the feed-consuming individual.

Thus, the model of optimal consumption presented in Equations (2)–(4) enables researchers to estimate ecological costs of feed consumption and to determine the efficiency of consumption and risk of individual's death.

Note that, technically, the model of feed consumption by insects perfectly matches consumption models in economic theory. In economics, the line described by Equation (3) is a line of budgetary constraints for a system consuming two products, and coefficients $a$ and $(b + 1)$ correspond to the costs of these products; the analog of Equation (8) is the so-called indifference curve, whereas $q$ is the utility of choosing certain prices [29,30].

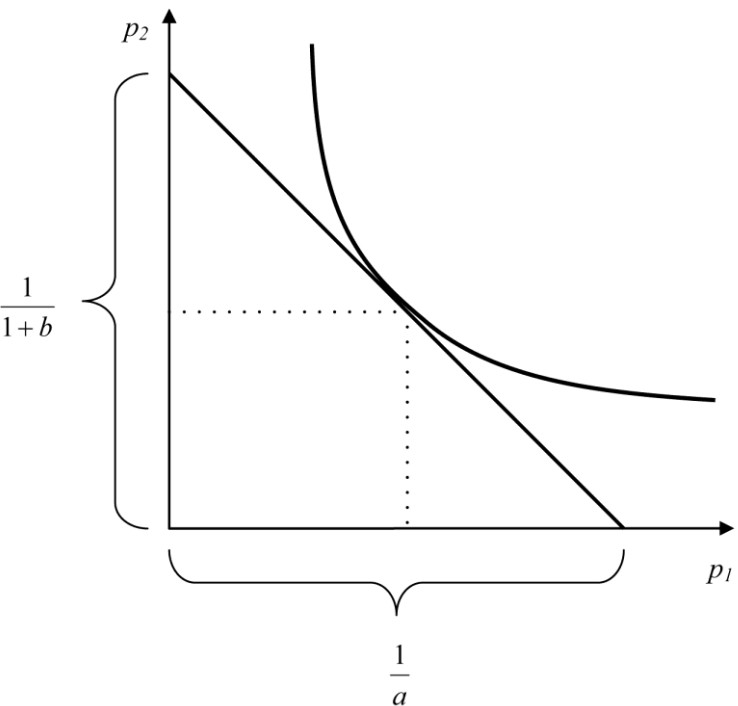

**Figure 2.** The relation between variables $p_1$ and $p_2$ within the feed consumption model.

## 3. Results

To verify model of optimal consumption (Equations (2)–(4)), experiments were conducted here regarding feeding (and weight gain) of spongy moth *L. dispar* caterpillars under laboratory conditions.

Figure 3 shows the relations between $p_1$ and $p_2$ (for spongy moth females) obtained in the experiments with feeding of caterpillars on birch leaves. This figure is comparable with the theoretical relation described above in Figure 2.

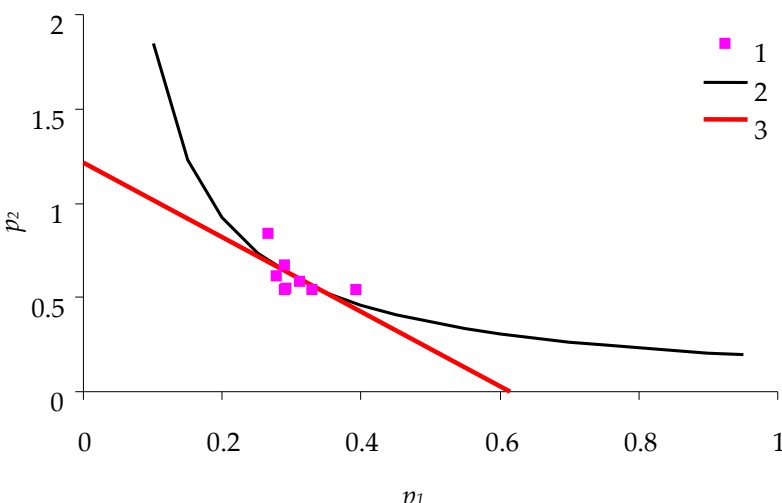

**Figure 3.** The energy balance of spongy moth females on birch: (1) characteristics ($p_1$, $p_2$) of individual caterpillars; (2) efficiency curve $q = p_1 \times p_2$; (3) line of budgetary constraints on feed consumption.

From the characteristics of nutritional balance, the following parameters of consumption efficiency of birch leaves by spongy moth caterpillars were calculated:

- Efficiency of feeding $q = 0.185$: ratio of caterpillar biomass to the weight of consumed feed.
- Cost of metabolic degradation of feed $a = 0.61$; cost of biomass synthesis $b = 1.215$.

- Optimum: point of intersection (0.31; 0.61) of hyperbola $p_2 = 0.185/p_1$ and line of consumption balance. As one can see, the caterpillars grow almost in accordance with this optimum.

Figures 4 and 5 show the consumption balance curves for females feeding on coniferous plants: pine and larch.

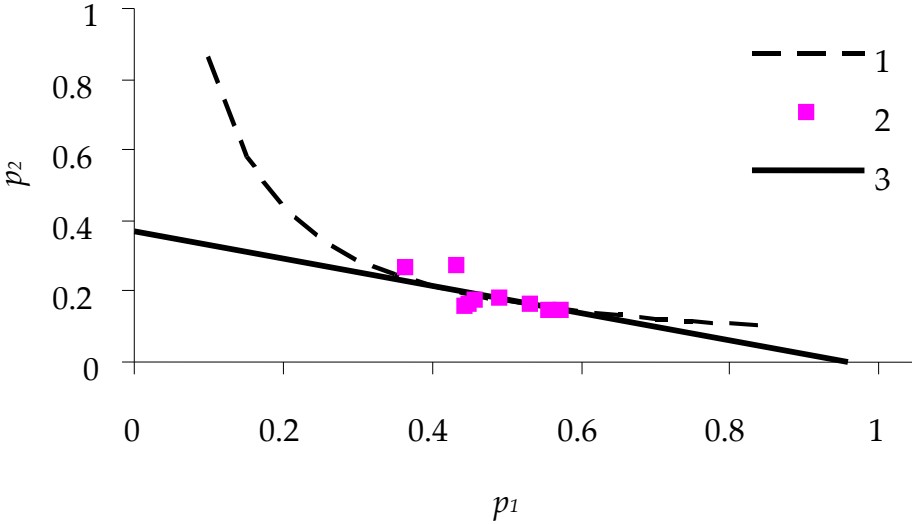

**Figure 4.** The energy balance of spongy moth females on pine: (1) hyperbola of optimal consumption; (2) data on caterpillars; (3) line of budgetary constraints.

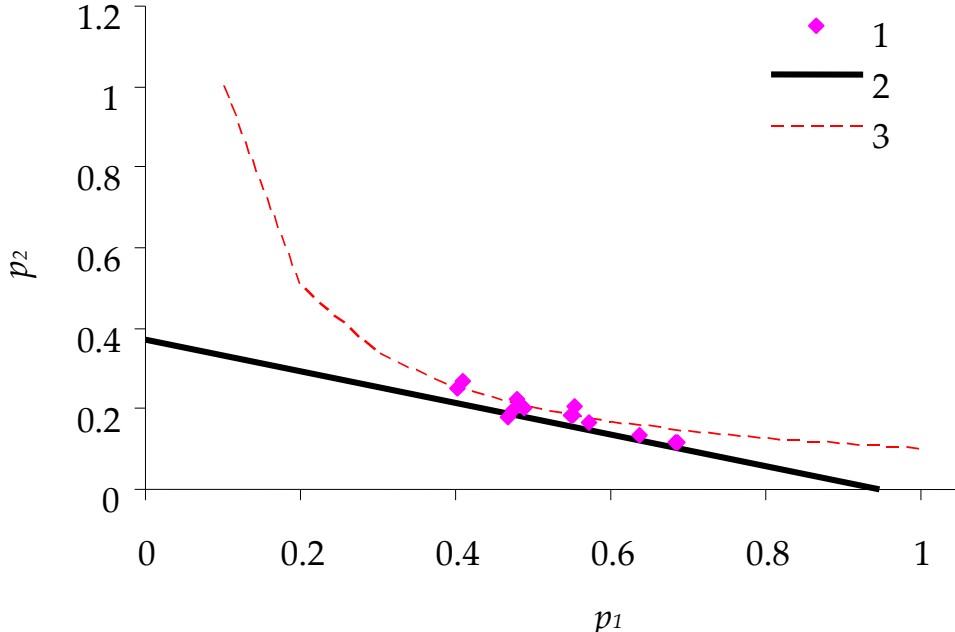

**Figure 5.** The energy balance of spongy moth females on larch: (1) data on caterpillars; (2) line of budgetary constraints; (3) hyperbola of optimal consumption.

Caterpillar feeding experiments allowed the calculation of the model parameters $q$, $a$, $b + 1$, $p_1$, and $p_2$ for different types of food plants. The calculated parameters are presented in Table 1. Since the sample size was small and the type of statistical distribution was not known, nonparametric tests (Mann–Whitney U-test) were used.

**Table 1.** Consumption model parameters for pupating females and males that fed on different types of feed. Errors refer to standard errors.

| Host Plant | Sex | Parameters of Consumption Model * | | | | |
|---|---|---|---|---|---|---|
| | | q $\pm$ SE | a $\pm$ SE | b + 1 $\pm$ SE | p1 $\pm$ SE | p2 $\pm$ SE |
| Birth | Male | 0.201 $\pm$ 0.008 | 1.43 $\pm$ 0.19 | 0.70 $\pm$ 0.11 | 0.35 $\pm$ 0.03 | 0.71 $\pm$ 0.105 |
| | Female | 0.185 $\pm$ 0.009 | 1.61 $\pm$ 0.19 | 0.82 $\pm$ 0.12 | 0.31 $\pm$ 0.01 | 0.61 $\pm$ 0.037 |
| Pine | Male | 0.083 $\pm$ 0.007 | *0.82 $\pm$ 0.05* | 3.57 $\pm$ 0.49 | 0.61 $\pm$ 0.035 | 0.14 $\pm$ 0.017 |
| | Female | 0.086 $\pm$ 0.0048 | *1.04 $\pm$ 0.05* | 2.70 $\pm$ 0.20 | 0.48 $\pm$ 0.02 | 0.185 $\pm$ 0.017 |
| Pine (for insects that died) | Males, females | 0.010 $\pm$ 0.005 | 0.52 $\pm$ 0.02 | 50.0 $\pm$ 0.01 | 0.963 $\pm$ 0.028 | 0.010 $\pm$ 0.007 |
| Larch | Male | 0.085 $\pm$ 0.004 | 0.84 $\pm$ 0.02 | 3.67 $\pm$ 0.23 | 0.60 $\pm$ 0.016 | 0.145 $\pm$ 0.01 |
| | Female | 0.100 $\pm$ 0.005 | 1.06 $\pm$ 0.05 | 2.70 $\pm$ 0.25 | 0.512 $\pm$ 0.033 | 0.224 $\pm$ 0.04 |

* $q$: efficiency of feeding (ratio of caterpillar biomass to the weight of consumed feed); $a$: cost of metabolic degradation of feed; $b + 1$: cost of caterpillar biomass synthesis, q: intersection point ($p_1$; $p_2$) between hyperbola and line of consumption balance. Significant differences (at $p < 0.02$) in the characteristics between males and females that consumed the same feed are italicized.

At the $p < 0.05$ level, differences in almost all parameters of consumption model between caterpillars that fed on different types of feed were significant, except for $p_2$ values of caterpillars that fed on larch and pine needles.

As shown in Table 1, feeding on birch leaves was the most efficient (for both females and males $q \approx 0.2$). For caterpillars feeding on larch needles, $q$ was ~0.09. For caterpillars feeding on pine and pupating at the same time, the costs of metabolic degradation of feed were close to those for larch. Nevertheless, for some caterpillars feeding on pine needles, the energy balance of feeding was minimal or even negative, and one-third of caterpillars died in the experiment.

Table 1 lists the parameters of feed consumption efficiency for caterpillars that fed on pine needles and died. As follows from these data, the feeding efficiency for such caterpillars was very low ($q = 0.01$), and there was a very high cost of biomass synthesis for this feed: $b + 1 = 50$. As displayed in Table 1, efficiency $p_1$ of feed metabolic degradation for males that survived and pupated when feeding on pine needles (as an unfavorable feed) was significantly higher than that for females.

On the other hand, efficiency $p_2$ of biomass synthesis was significantly higher for females than for males. Eventually, this imbalance led to very similar values of feeding efficiency $q$ between males and females. In this context, parameters of feeding efficiency $p_1$, $p_2$, and $q$ for 10 caterpillars feeding on pine needles (subsequently died) were significantly lower than those in surviving caterpillars.

For caterpillars feeding on larch needles, feeding efficiency was close to the levels seen earlier in independent experiments with feeding (and weight gain) of spongy moths on larch [8]. According to the results of present work, for male spongy moths fed on larch needles, $q = 0.129 \pm 0.01$, and, for females, $q = 0.095 \pm 0.015$. At the same time, both the costs of metabolic degradation of feed and the costs of caterpillar biomass synthesis were quite high.

It should be mentioned that the efficiency of feed consumption $q$ is not linked with the growth rate of the caterpillar weight. In Figure 6, on a semilogarithmic scale ($t$, ln $m$), the growth curve (typical for these experimental conditions) of weight $m$ of a spongy moth caterpillar is shown, from the beginning of the feeding experiment to pupation.

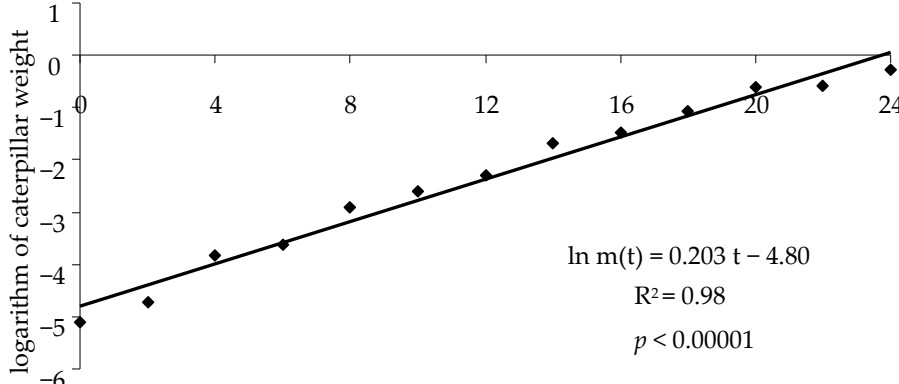

time *t* from start of experiment, days

**Figure 6.** Dynamics of caterpillar weight growth (female No. 11, host plant species: Siberian larch).

The weight growth equation for caterpillar can be written as follows:

$$\ln m(t) = k + vt \tag{10}$$

where *k* is the logarithm of caterpillar weight at time *t* = 0 (start of experiment), and *v* is the rate of caterpillar weight growth.

For all caterpillars that fed on Siberian larch needles, Figure 7 depicts the relationship between weight growth rate *v* and feed consumption efficiency *q* for females and males.

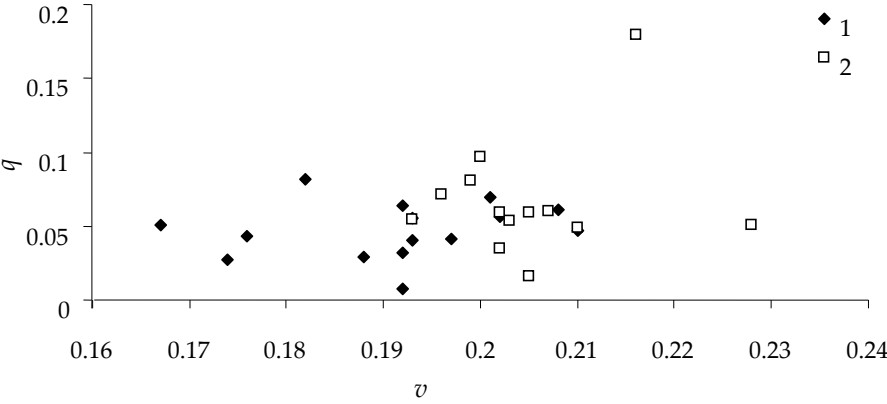

**Figure 7.** Relations between weight growth rate *v* and efficiency *q* of feed consumption for male (1) and female (2) spongy moths.

As can be seen in Figure 7, the weight growth rate for male spongy moths was less than that for females. This species has well-pronounced sexual dimorphism (as reflected in the scientific name of the species) and the final weight of pupae; hence, the rate of weight gain is one of the criteria for sexual differences [31]. Nonetheless, there was no significant relationship between the growth rate of caterpillars (females and males) *v* and the efficiency of feed consumption *q*; it can be concluded that feeding efficiency is not linked with sexual dimorphism of the species, although variation of this trait is quite wide.

An important indicator of viability of a population is fecundity *F* of females. Our analysis indicates that there was a linear relationship between feeding efficiency *q* of the caterpillar and its fecundity *F* expressed in the weight of eggs produced by the caterpillar (Figure 8).

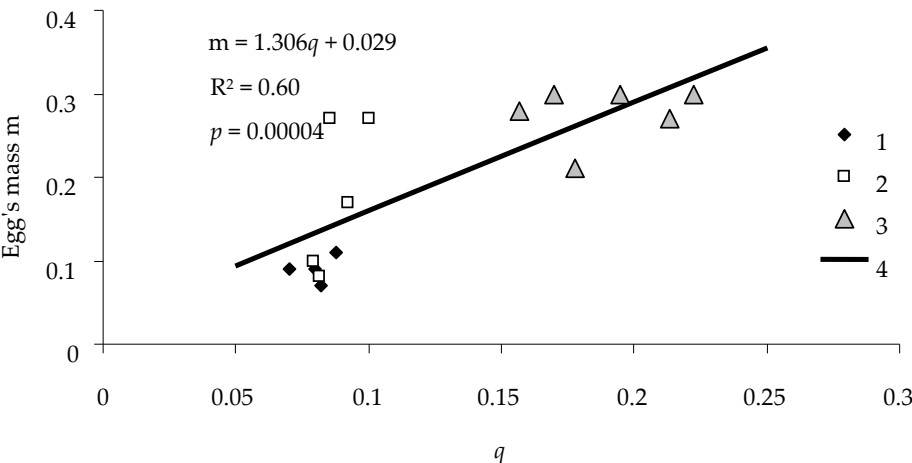

**Figure 8.** The relationship between caterpillar feeding efficiency *q* and caterpillar fecundity *F* (weight of laid eggs) for insects reared on all types of feed. Host plant species: (1) pine; (2) larch; (3) birch; (4) regression line for caterpillars feeding on all the host species.

As displayed in Figure 8, the relationship between feeding efficiency *q* and fecundity *F* was significant, and the coefficient of determination $R^2$ for the correlation between *q* and *F* was 0.60. The responsiveness $\frac{\partial m}{\partial q} = 1.306$ of the weight of eggs to a change in the efficiency was positive (i.e., with an increase in *q*, *m* also increased quite strongly).

## 4. Discussion

How do the models in Equations (2)–(4) proposed in this paper differ from the standard balance model in Equation (1)? In the proposed model, two equations were introduced to describe energy balance (in mass units). This allows, according to feeding data, unambiguously evaluating the distribution of energy to various processes during feed consumption. The difference between the balance Equation (2) and the traditionally used Equation (1) is the presentation of metabolic costs in the form of two independent components—the cost of preparing feed and the cost of proper metabolism. Such a division in Equation (1) is impossible, because, in this case, two independent variables appear and it is impossible to find them from one equation (in fact, just the total accounting balance can be obtained). When introducing a two-component representation of metabolic costs, one more equation is needed for calculation. To this end, an equation can be introduced on the basis of the optimization principle in Equation (4). According to this model, during the process of food consumption by insect caterpillars, the maximum efficiency of food consumption is ensured (ceteris paribus). In this case, according to the experimental data, it is possible to calculate new nutritional characteristics—the feed conversion costs and the metabolic cost of synthesizing caterpillar biomass. Comparison of feeding data for caterpillars with the proposed model indicates a very good fit. In this case, the prices of assimilation and consumption characterize the quality of feed. For birch leaves and larch needles, food prices differ significantly, and the prices of pine needles for insects are so high that, for many individuals, a positive energy balance is not provided when using food for individual growth.

Calculations with experimental data indicate the possibility of using the optimization principle in Equation (4) to describe the feeding process. In addition, the models in Equations (2)–(4) explain the death of caterpillars when feeding on certain food (in our experiment, some caterpillars died when feeding on pine needles). The balance Equation (1) always has a positive solution for metabolic costs; accordingly, there are no reasons for the death of individuals. When metabolic costs are divided into two components and consumption prices are introduced, solutions are possible at which *q* < 0. This means that the use of food with such ecological prices does not allow caterpillars to balance energy,

which leads to death of an individual, analogous to bankruptcy in the economy when prices rise.

The introduction of new characteristics to describe metabolic costs makes it possible to introduce an idea about insect feeding strategies that is impossible according to the model in Equation (1).

Previously, one of the authors of this article analyzed the feeding efficiency of *Aporia crataegi* L. caterpillars (Lepidoptera, Pieridae) [15]. For this species, which also manifests outbreaks of mass reproduction, the efficiency of assimilation of feed $p_1 \approx 0.90$ is very high (much higher than that for spongy moth with any type of feed), but the efficiency of biomass synthesis $p_2 \approx 0.08$ is much lower (substantially lower than that for spongy moth). At these values of $p_1$ and $p_2$, feed consumption efficiency $q$ is ~0.072 for both females and males. Comparing the findings from experiments with spongy moths and *A. crataegi*, apparently, we can distinguish at least two strategies for feed consumption. The first strategy is characterized by high efficiency of metabolic degradation of feed but low efficiency of biomass synthesis ($p_1 > p_2$). When the second strategy is "chosen", there is an emphasis on the efficiency of biomass synthesis ($p_1 < p_2$). It can be concluded from Table 1 that the choice of either strategy is determined not only by the species of consumer insect but also by the species of the host plant. When feeding on birch, spongy moths tend to maximize the efficiency of caterpillar biomass synthesis ($p_2$ = 0.6–0.7 > $p_1$ = 0.31–0.35). During feeding on pine, the opposite is true: $p_1$ = 0.48–0.61 > $p_2$ = 0.14–0.185. The strategy for feeding on larch needles is similar to that on pine needles. Nevertheless, for some caterpillars, at the maximal efficiency of feed metabolic degradation, when $p_1$ approaches 1.0, all the energy received from feed is spent on its metabolic degradation; therefore, the growth of caterpillar biomass becomes impossible, as in the case for some spongy moth caterpillars feeding on pine needles (Table 1). Obviously, the situation when both $p_1$ and $p_2$ are close to 1.0 (and consequently $q \to 1.0$) is impossible (in fact, the first law of thermodynamics prohibits this situation).

Thus, knowing feeding efficiency for a certain type of feed, a researcher can estimate the expected fecundity of the insects in question. On the other hand, the individuals feeding on larch are probably under the influence of additional factors that positively affect the fecundity of insects. The importance of feeding efficiency also lies in the fact that it allows a researcher to assess the suitability of a feed for nutrition and the risk of death of caterpillars owing to an energy imbalance in nutrition. According to our experiments, $q$ = 0.01 leads to the death of these insects.

## 5. Conclusions

The proposed model of feed consumption makes it possible to separate processes of metabolic degradation of feed and insect biomass synthesis. This approach enables investigators to identify different feeding strategies depending on the type of feed and possibly on the habitat of the insect species. An important task is to evaluate the relationship between the abovementioned "costs" of feed and the risk of population outbreaks. With an increase in the efficiency of feed consumption, the fecundity of individuals grows, which directly affects reproduction coefficient of population; consequently, a decrease in feed "costs" can cause an increase in population size and outbreaks.

How can the efficiency of feed consumption change with climatic shifts and with expansion of the geographic range of forest insects to the north? It follows from Table 1 that, for insects to survive, feed consumption efficiency $q$ must be greater than 0.08. Nonetheless, it is not clear how $p_1$ and $p_2$ will be linked during the decrease in heat supply with migration of insects to the north. For example, it is known that plant phenology (which is certainly different in northern regions) can influence the efficiency of spongy moths' assimilation of various plant species [32]. Caterpillars of the spongy moth in the marginal northern and central populations react differently (in terms of the rate of development) to dissimilar levels of heat supply and are differently susceptible to changes in feed depending on the feed history [19]. It can be expected that, when host plant species change, the efficiency

of feeding may differ from the value obtained in model experiments. Nevertheless, our experimental data based on nutrition economics indicate only low potential for the development of spongy moth on pine, although, if all the caterpillars reach an efficiency of 0.08 for the consumed feed, they can survive and produce offspring. In actuality, spongy moth abundance is unlikely to grow when feeding on pine.

One important clarification is that we carried out this study on a spongy moth population belonging to a European biotype (earlier known as the European gypsy moth, EGM), while there is an Asian biotype (Asian gypsy moth, AGM) which shows differences in the preferred host plants [33]. Because of the high level of phenotypical polymorphism in *L. dispar* populations, we recommend caution in applying values of nutrition efficiency presented in our study to another *L. dispar* biotype.

It is worth mentioning that the examined model of insect feed consumption and the model of consumption of two goods well known in economics [30] are basically identical, and there seems to be a systemic similarity of consumption processes between animals and *Homo economicus*.

**Author Contributions:** Conceptualization, V.M. and V.S.; methodology, V.M. and O.T.; experiments, Y.A., E.O. and S.P.; data curation, V.S., Y.A. and A.K.; writing—original draft preparation, V.S.; writing—review and editing, V.M. and A.K. All authors have read and agreed to the published version of the manuscript.

**Funding:** This study was supported by the Russian Science Foundation (grant Nr 20-64-46011) for Analytical Tasks and by the Federal Fundamental Scientific Research Program for 2021–2025 (FWGS-2021-0003) for wet lab studies.

**Institutional Review Board Statement:** Not applicable.

**Informed Consent Statement:** Not applicable.

**Data Availability Statement:** Not applicable.

**Conflicts of Interest:** The authors declare no conflict of interest.

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
