# Peer review of "Economics of a Feeding Budget: A Case of Diversity of Host Plants for Lymantria dispar L. (Lepidoptera) Feeding on Leaves and Needles"

_diversity, doi:10.3390/d15010102_

Round 1
Reviewer 1 Report
Introduction - some statements are poorly supported by References. Examples: rows 36-36, on the factors of fecundity; rows 43-44, on the relation between food and weight of individuals; rows 48-49, assessment of feed quality by energy balance of consumption. The metabolic processes are mentioned rather superfluously, the phytochemical relations of the food quality (tannin content) are unconsidered, despite of the numerous References on this topics. Rows 109-110,
Studied species: The statement on Betula pendula as main food plant of L. dispar in "endemic part" of the range is a simple error, see the References of Serbian authors e.g. Milanovic et al. Eur. J. Entomol. 111(3): 371–378,
Methods: According to my opinion it is a serious deficience of methods (rows 140-142) that the tannin content of leafs and needles was not measured. Many references demonstrate that the tannin content and its variation in time and space are relevant factors of growth and ferility and, therefore for population dynamics of L. dispar.
Row 167- The two additive components of the equation (2) are not independent from each other (and therefore cannot be additive!), since these depend from the quality, i.e. chemical composition of the food. In the rows 173-174 the same error is repeated.
Results: the presentation of Results (and graphs) are elegant, indeed. However the simplified presumptions of the models seem for me rather problematic (see above).
Rows 321-326 - Lymantria dispar and Aporia crataegi are compared. OK, both are common or even pest species e.g. in South Siberia (my personal experiences, too), however the real background of the difference between them is masked by the simple circumstance that larvae of L. dispar are highly "protected" against the harmful effect of tannin (s. publications of L. dispar feeding on oak!), while the caterpillars of A. crataegi are feeding mostly on Rosaceae nearly without any tannin content. Therefore the whole comparison is going in wrong direction.
Conclusions. Rows 345-346. The basic problem is that the insect biomasse synthesis may be the (more or less direct) consequence of the success of the metabolic degradation of the food. It means that some chemical factors of the food quality should not be neglected. This is the main deficience of the whole study! Therefore the model "an sich" seems to be rather problematic for me! Rows 160-161. The generalisation, formulated here, seems for me problematic! The western and eastern Eurasiatic populations of L. dispar are genetically deeply divergent (e.g. in the lavel of sex dimorphism. see: "flying" females in South Siberia and Mongolia). These Eastern pop-s are much more adapted e.g. to feeding on birch than the pop-s of the European deciduous forest zone which have much more food plants but preferably Quercus spp.. These aspects should not be neglected neither isn the Discussions nor in the Conclusions.
Two technical details:
Lymantria dispar is usually mentioned in the Ref.-s as "Gypsy Moth". I do not see any reasonable ground to change it. "Spongy moth" is a wrong traduction of the German "Schwammspinner", any connection with Porifera. Even "wooly moth" would be better.
Taxonomic position: according to relevant taxonomic references: Erebidae, subf. Lymantriinae (and not Lymantriidae).

Author Response
Dear Reviewer,
Thank you very much for your comments. Summarizing the criticism of first referee we could conclude that one of our serious omission in this MS is the absence of study of leaf/needles secondary compounds (primary tannins) within this study. We know about the effect of secondary compounds on many L .dispar parameters, including survival, potential productivity, actual and potential resistance against biotic factors such as parasites ets (see Martemyanov et al., 2010, 2012a,b, 2015a,b, Chernyak et al., 2016). Why referee try to focus on polyphenolic compounds (tannins) it is not clear for us because there is excellent study of Raymond Barbehenn and Peter Constabel who summaries the effect of condensed tannins (proanthocyanidins if you wish) and hydrolysable tannins on herbivores in they review article and clear indicate that L. dispar is tannin tolerant species. Many experimental studies cited within that article clear indicate this. We would agree with this conclusion because most of phenolic compounds in silver birch does not correlated with L. dispar life history traits (Martemyanov et al, 2012, Journal of chemical ecology). The same result was shown when we studied the seasonal dynamics of birch leaf chemistry: content of condensed tannins were not changed significantly during first ten days of leave development (Chernyak et al., 2016) while life history traits of L. dispar were negatively effected by those aged leaves (Martemyanov et al., 2015, PLoS ONE, Asynchrony between…..). That is most important, we do not find such association between seasonal dynamics of easily oxidized phenolics (see excellent method developed by Juha-Pekka Salminen) – most reactive phenolics providing negative effect on folivores by generating of rective oxygen species at alkaline pH. Basin on another studies (Martemyanov et al., 2012, Arthropod-Plant interactions) we conclude that such lypophilic compound as phenolic aclycones and triterpnoids (Martemyanov et al., 2015, PLoS ONE, Leaf Surface Lipophilic Compounds….) are more important for chemical defense of birch against Lymantria dispar. Thus we do not think that we need additionally present the data about quantitative presence of tannin in larvae diet. Moreover conifer species (pine and lurch) used in considering study seriously defer in chemistry from birch. Right now we are submitting the study about phytochemistry of that trees in another journal. Thus if we need to focus on leaves/needles secondary compounds we need wider looks chemicals classification. What is do important for the current MS is that how L. dispar will response on different host tree species consumption and we provide these data. Of course this data include the reaction of gypsy moth (spongy moth in modern manner) on the set of barriers of constitutive resistance of studied tree species (primary compounds, secondary compounds, physical-chemical barriers such as trichrome, functioning of polyphenol oxidase, which increase the negative effect of phenolics, etc.). We do not have primary aim in this study to reveal the mechanism of chemical defense between compared species of trees. We try to estimate the economics of feeding of folivore invading to the north forests by compared trees species. This is our position in this question and our following comments will base on this statement and providing above arguments.
Also it seems that referee consider the initial MS (MS was already improved accruing to academic editor recommendations), so some comments is already done.
- Line 17. Unfortunately, last year meeting of ESA (Entomological society of America) accept new common name that now confuse reader. As we use English name we need to follow by corresponding spelling. We add the mention of previous common name in the abstract for easier searching procedure, thank you for the comment.
- Line 25. We do not understand why this text is in yellow but if this comment belong to conclusion, see our response in corresponded section.
- Line 36 this is general statement in ecology, but we add the reference on basic ecological book
- Line 37-38 we add the reference for this statement which show this correlation for studied species.
- Line 43-44 please read following sentence in the text which indicate the importance of quality. Our logic is writing from quantity to quality, why not.
- Line 48-50 Please find following sentence with examples of this statement
- Line 62 Tree constitutive defense (we do not consider induced defense within this study) is more complicated mechanism than tannin concentration or even whole chemical defense of tree (see our statement above). We consider in our experiment leaves/needle quality as mixture (or package) of features of certain tree species and then explain our findings in this context. We do not want to show the mechanism of constitutive defence for each tree species. It was done by Kenneth Raffa and co-authors for lurch, some another American or German or Serbian teems for deciduous trees.
- we do not understand what referee mean
- Line 76 yes we agree that cited studies are restricted by language knowledge that is why now we are presenting current study in international journal for wider auditory.
- Line 104 thank you for the comment, we have corrected it
- Line 109 thank you for the comment, we have clarified the part of range to avoid misunderstanding because of wide trophic specialization of L. dispar.
- Line 121 thanks for careful reading! we have corrected
- Line 140-142 We careful justify our position in this question in first paragraph and in the response # 5. If it will not convince referee, we will address to academic editor to consider arguments of referee vs arguments of authors and provide the independent decision .
- Line 166-167. The variables in equation (2) are not directly related to each other by their definition, the identification of their possible relationship is associated with the hypothesis of decomposing metabolic costs into two components.Testing this hypothesis and using a non-linear programming model to calculate the model variables seems to be a new result and allows us to compare the processes in the Human economics and food consumption by insects.We would say that this fact can be understood as a systemic similarity of biological processes in the biosphere.
- Line 173 нужно подумать in expression (2) for the variables p1 and p2, it is precisely the nonlinear model that is proposed.
- Line 321 Again here. Mechanisms of plant antibiosis is wider than presence/absence tannins in foliage. Both species are polyphagous lepidopterans so the comparison is possible. The strategy of feeding balance will depend on the mechanism of plant antibiosis and mechanism of overcoming this antibiosis by consumer (i.e. arm race ). We compare the strategies of these arm races by two species of consumers. We are not trying to understand what biochemical differences lead to different reactions of two species of insects by host plant. We simply show that different feeding strategies are possible for different species. The choice of Aporia crataegi is due to the fact that one of the authors of the presented work (О.Т.) studied the feeding of this species and we obtained data for comparison
- Line 345 the model include the effect of such factor as tree chemical defense. We do not have the purposes for this study to calculate the separate cost for secondary compounds metabolism (diterpenes in the case of lurch and triterpenoids in the case of Silver birch).
It was not our intention to present studies on the biochemistry of insect nutrition. The purpose of our work is much more modest - to show how non-linear programming methods allow us to describe concepts such as "prices" of feed, structurally similar to the concepts of prices in economics. It is impossible to demand full coverage of all problems from one article.
- Line 359 Unfortunately we do not understand referee here. All we would like to conclude is that values get in our lab conditions should be extrapolated on natural populations WITH COUTION because of the reasons mentioned in referred studies. How it is related with divergence of AGM and EGM it is not clear for us. By the way according to resent studies of Michel Cusson group, EGM is spread until Yenisei river and does not divided by Ural mountings as suggested earlier. So in our study we used EGM, although population genetics of L. dispar is more complicated than dividing on EGM and AGM (Picq et al., 2023, in press in Evolutionary applications).
All corrections are visible in the MS article text.
Sincerely, Authors.
Reviewer 2 Report
The manuscript is written nicely briefing the importance of modeling in understanding process of consumption and metabolism. Few suggestions for further improvement of MS is, in the title, Line-3: 'or' may be replaced with 'and'.
Line 17- delete 'these'
Line 188 and 189 - It is not clear how the question is obtained.
Author Response
Dear Reviewer,
Thank you very much for your comments, we have taken into account and corrected them all.
Line-3: replaced with 'and'.
Line 17-delete 'these'
Line 188 and 189 - corrected for "algebraic transformations". The energy E is explicitly excluded from equation.
All corrections are visible in the MS text of the article.
Sincerely, Authors.
Round 2
Reviewer 1 Report
Please see the attachment.

Author Response
Dear Editor, referee
We carefully study the comments. Thank you for careful work! As we understand almost all issues were justified. One unaddressed issue is the last paragraph of referee comment where he/she describe the importance of differences between two biotypes of L. dispar (i.e. AGM vs EGM). We guess that it is important clarification because of so huge phenotypical variation of spongy moth populations. We add corresponding paragraph in the conclusion section alerting about possible effect of this phenomenon. Hope we have fixed asking issue. We cite related reference. We do not want to city presented by referee reference from Scientific Reports journal because the final idea and that is more important, got results from that article are far from our statement given in improved version. One of co-author of our MS being the referee of that MS, so he studied that data in detail. We provide more related reference.
Regarding the comments from the academic editor, we have done it already. Possibly there is some technical problems in the synchronization of way of correspondence of MS versions (i.e. via electronic system vs. via e-mail), please check it.
So we hope that current version of improved MS fix all mentioned issues.
Once again thanks for the criticism and Happy New Year to all involved to the work on this MS!